# Effects of Diet and Supplements on Parameters of Oxidative Stress, Inflammation, and Antioxidant Mechanisms in Patients with Chronic Renal Failure Undergoing Hemodialysis

**DOI:** 10.3390/ijms252011036

**Published:** 2024-10-14

**Authors:** Anna Bogacka, Maria Olszewska, Kazimierz Ciechanowski

**Affiliations:** 1Department of Commodity Science, Quality Assessment, Process Engineering, and Human Nutrition, West Pomeranian University of Technology in Szczecin, 71-459 Szczecin, Poland; 2Departament of Medical Chemistry, Pomeranian Medical University in Szczecin, 70-111 Szczecin, Poland; 3Clinical Department of Nephrology, Transplantology and Internal Medicine, Pomeranian Medical University in Szczecin, 70-111 Szczecin, Poland; kazimierz.ciechanowski@pum.edu.pl

**Keywords:** hemodialysis, nutritional state, oxidative stress, MDA, cytokines, CAT, SOD, GSH-Px

## Abstract

The prevalence of chronic kidney disease (CKD) worldwide increases as the population ages. The progression of the disease increases the risk of complications and death and leads to end-stage renal failure, requiring renal replacement therapy. Despite the positive effect of hemodialysis (HD), patients are at risk of developing malnutrition, inflammation, oxidative stress, or cardiovascular disease, which worsens quality of life and can lead to organ dysfunction. The occurrence of the mentioned disorders depends largely on the diet, so changes in diet composition are an important part of the treatment of kidney disease. This study aimed to evaluate the effects of a balanced diet on some parameters of oxidative stress, immune response, and nutritional status in patients. This study included 57 HD patients (19 women and 38 men). In all of them, nutritional status and diet were initially determined, and then, they were divided into six groups, which received different diets and supplements. Serum levels of albumin, total protein, MDA, and the cytokines Il-1, IL-6, IL-8, TNF-α, and IL-10 were determined, and the activity of the enzymes such as CAT, SOD, and GSH-Px were determined in erythrocytes by spectrophotometry. Based on the results of BMI, albumin, and total protein, it can be concluded that a well-balanced diet can reduce weight loss. This study shows that a well-balanced diet can reduce the secretion of pro-inflammatory cytokines, and ensure the normal activity of antioxidative enzymes in the blood of HD patients.

## 1. Introduction

The prevalence of chronic kidney disease (CKD) increases as the world’s population ages and the number of patients with diabetes and hypertension rises. According to a report by Fresenius Medical Care, 5,071,000 people worldwide were treated for kidney failure in 2023, including 3,628,000 with hemodialysis and 444,000 with peritoneal dialysis (PD) [1]. In 2022, there were 20,198 people on dialysis (HD + PD) in Poland [2]. The optimal therapy for these patients is kidney transplantation, but not all patients meet the criteria necessary for surgery, and the waiting time for kidney transplantation is long. Therefore, it is necessary to further develop dialysis therapy as it can be a chance for long-term survival of patients with chronic renal failure [3].

In patients with chronic renal failure (CRF) undergoing hemodialysis, there is a complex biochemical process in which the positive effects of treatment are simultaneously associated with some negative effects. Although hemodialysis is a key life-saving treatment, patients often experience increased production of reactive oxygen species (ROS), which can lead to cell damage. The neutrophils of dialysis patients are more active in producing ROS compared with the neutrophils of healthy individuals [4,5]. Such an increase may be caused by increased oxidative stress and the action of pro-inflammatory cytokines such as interleukin-1 (IL-1), tumor necrosis factor-alpha (TNF-α), interleukin-6 (IL-6), and interleukin-8 (IL-8), which are produced when blood comes into contact with the dialysis membrane [6,7].

Free radicals and ROS under in vivo conditions cause the oxidation of polyunsaturated fatty acids. Oxidation results in the formation of fatty acid peroxides, which further degrade to keto- and hydroxy derivatives. Their further degradation leads to the release of shorter fragments and aldehyde derivatives. One of these is malondialdehyde (MDA, malondialdehyde), an indicator of the lipid peroxidation process [8]. Lipid peroxidation products react with the thiol groups of proteins and with the amino groups of proteins, lipids, amino sugars, and nitrogenous bases that make up nucleic acids to form Schiff bases [9]. Aldehydes damage cell membranes, causing depolarization, resulting in loss of integrity. In in vitro studies, MDA has been shown to inhibit membrane enzyme activities and indirectly participate in protein, DNA, and RNA synthesis [10].

Increased free radical production and activation of the antioxidant enzyme system can lead to inflammation and further tissue damage. These include catalase (CAT), superoxide dismutase (SOD), and glutathione peroxidase (GSH-Px). These enzymes are essential components of defense against increased levels of oxidative stress [11].

Catalase converts hydrogen peroxide to oxygen and water and oxidizes substances such as methanol, ethanol, phenols, and nitrites [12]. People with low CAT activity have a higher risk of developing oxidative stress-related diseases, including dyslipidemia and related diseases [13]. Superoxide dismutase is a metalloenzyme that affects the breakdown of superoxide anion, which promotes, among other things, the formation of lipid radicals. These radicals destroy lipid membranes, which, in turn, leads to reduced fluidity and altered permeability. SOD converts the superoxide anion into hydrogen peroxide, which is then converted to water and oxygen by glutathione peroxidase (GSH-Px) and catalase (CAT) [14]. Glutathione peroxidase (GSH-Px) is a type of antioxidant enzyme responsible for catalyzing the breakdown of lipid peroxides. Both hydrogen peroxide and organic hydroperoxides exhibit high GSH-Px activity [15].

The long-term effects of these processes can be particularly detrimental, leading to increasing cachexia and dysfunction of various organs. Cachexia, characterized by loss of muscle mass and weakness, is a consequence of chronic disease and inflammation, which further affects the quality of life in patients with CRF [16,17].

It is, therefore, important to apply measures to minimize oxidative stress and control inflammation in HD patients. This may include the use of antioxidants, dietary modification, and optimization of dialysis procedures.

A non-pharmacological intervention is the selection of an appropriate diet to meet the needs of the patient’s body. The purpose of properly selected (adequate) dialysis, as well as a properly balanced diet, is to provide optimal amounts of all nutrients to protect against the development of malnutrition, reduce the accumulation of nitrogenous metabolic products in the body, give the patient a sense of freedom, and improve the quality of life [18,19]. An individually tailored diet can slow the development of kidney damage and prevent metabolic disorders that occur in the course of chronic kidney disease (hyperkalemia, hyperphosphatemia, hypocalcemia) [20]. In addition, a properly composed diet should facilitate the control of blood pressure, hyperglycemia, and lipid disorders that accelerate the progression of CKD [21].

CRF patients are particularly predisposed to the development of nutritional status disorders. The causes include dietary restrictions in the pre-dialysis period, loss of appetite, and increased catabolism [22].

Several papers have shown that a significant proportion of patients start dialysis treatment with symptoms of malnutrition, while in others, it appears later [22,23,24]. The consequences of nutritional deficiencies (energy, protein, vitamins, minerals) in dialysis patients are numerous, including symptoms of uremia, decreased protein synthesis, increased catabolism, metabolic acidosis, chronic inflammation, and insulin resistance [25]. Complications can be reduced by proper nutrition. The diet should contain an adequate amount of calories; moreover, it should be quantitatively and qualitatively balanced. Vitamins and bio-elements are particularly important components [26].

Changes in dietary composition are a key component of renal disease management in all periods of chronic kidney disease (CKD). Dietary recommendations include the supply of essential nutrients: protein, carbohydrates, fats, vitamins—especially antioxidant vitamins—and minerals such as phosphorus, potassium, sodium, calcium, and fluids [27].

According to current guidelines, it is recommended that HD patients who are metabolically stable consume 25–35 kcal/kg b.w./24 h, depending on age, gender, physical activity, and comorbidities. Carbohydrates should provide 50–60% of the total energy requirements, fats a maximum of 30%, with saturated fatty acids as little as possible. The main source of energy should be complex carbohydrates (grains, macaroni, breads) and vegetable fats. It is recommended to limit the intake of simple sugars. In HD patients, a dietary protein intake of 1–1.2 g/kg b.w./24 h is recommended. As for the type of protein (vegetable or animal), it should be of high biological value, while KDOQI does not give clear recommendations in this regard [28]. Daily potassium intake of HD patients should not exceed 2000 mg to prevent hyperkalemia. In addition, it depends on the concentration of potassium in the blood. Potassium from meat and dairy products is absorbed more easily from the gastrointestinal tract than from fruits, vegetables, legumes, nuts, and seeds. A plant-based diet is, therefore, possible during hemodialysis [29]. Sodium intake should be limited to 2000 mg/day. Dialysis patients require an increase in dietary calcium, up to 1000–1500 mg/day, and limiting phosphorus to 1200 mg/day. Products rich in protein contain significant amounts of phosphorus. Limiting phosphorus intake is difficult to achieve without restricting protein intake. HD patients are, therefore, given calcium carbonate salts that bind phosphorus in the gastrointestinal tract to prevent hyperphosphatemia. The drug must be administered immediately before meals [30]. Loss of vitamins during HD, dietary restrictions, absorption, and metabolic disorders can cause vitamin deficiencies, especially water-soluble ones. Therefore, vitamin supplementation is recommended during renal replacement therapy [28]. A dose of 90 mg of vitamin C per day for men and 75 mg/day for women is recommended as the most appropriate in HD patients. Supplementation with B vitamins and folate should depend on the clinical condition of patients. Also, administration of vitamin E, which is an antioxidant, seems to be beneficial, but a safe dose for routine administration to HD patients has not been established. Vitamin A supplementation, on the other hand, is not recommended. In dialysis patients, high serum concentrations are observed due to an increase in retinol-binding protein (RBP), a decrease in protein catabolism, and a lack of removal during dialysis. Given the probable involvement of vitamin A and RBP in exacerbating uremic pruritus, vitamin A supplementation is unfavorable [31].

The amount of fluid intake must be controlled. It depends on the volume of urine passed. Fluid intake should be limited to an average of 500–800 mL/day (including those contained in products) plus the amount of urine excreted. Increasing fluid intake can result in significant weight gain. The increase in weight between dialysis should not be greater than 2–5% of the patient’s dry weight [32].

This study aimed to evaluate the effects of a balanced diet enriched with antioxidant products on some parameters of the antioxidant system (CAT, SOD, GPx), oxidative stress (MDA), immune response (IL-1, IL-6, Il-8, TNF-α, IL-10), and the course of cachexia in HD patients.

## 2. Results

### 2.1. Results of Anthropometric and Biochemical Parameters

#### 2.1.1. BMI

Table 1 shows the BMI values of the patients in each group during the dietary therapy. The value of the index in the study groups did not change significantly during the 12 months of the study.

Group I had the highest BMI regardless of the month of measurement. At the start of the study, the patient’s weight was at the upper limit of normal BMI and increased in the subsequent period. Lower BMIs were observed in Groups II, III, IV, and V. Accepting a BMI range of 18.5–24.9 as normal and 25.0–29.9 as overweight, there were no decreases in weight or excessive growth during the follow-up in the groups fed under dietitian supervision. In contrast, Group 0, being on a home diet, showed a decrease in BMI at the 6th month of measurement.

Therefore, it has been shown that the use of a well-balanced diet can affect the nutritional status of HD patients, either through a slight increase in body mass index or by maintaining baseline values.

#### 2.1.2. Albumin Concentration

One of the biochemical exponents of increasing malnutrition may be reduced values of albumin concentration (Table 2). The norm was a concentration of 35–50 g/L, values of 35–39 g/L were taken as the lower limit of the norm, and 40–46 g/L as the middle limit.

At the beginning of the study, the lowest plasma albumin content was observed in Groups 0 and I, higher content in Group V, and the highest in Groups II, III, and IV.

Statistical analysis showed significant differences between the following groups: 0 and II, III, IV, between I and II, III, IV, and between IV and II and V. Albumin concentration in Group I was statistically significantly lower than in Groups II, III, and IV, while in Group IV, it was significantly higher than in Groups 0, I, II, and V. The mean albumin concentration in Group 0 was significantly lower than in Groups II and III, while in Group III, it was significantly higher than in Group I.

In Group 0, albumin concentrations fell to the lower normal limit after 6 months of the study. This situation did not occur in the other groups that followed the principles of proper nutrition where an increase in albumin concentration was evident. Thus, it can be concluded that a proper diet is conducive to maintaining good nutritional status.

#### 2.1.3. Protein Concentration

The serum total protein content of the patient groups is shown in Table 3, with statistically significant differences between each month in which the determinations were made. The concentration of total protein in the blood of properly fed hemodialysis patients increased significantly over time, although it decreased in Group 0.

Using ANOVA analysis, statistically significant differences were shown between Groups 0 and II, III, IV, and V, between Groups I and III, IV and V, and between II and III, IV and V. The mean total protein contents of Groups III, IV, and V were statistically significantly higher than those of Groups 0, I, and II. Group II had a significantly higher total protein content than Group 0 (Table 3).

### 2.2. Results of the Determination of the Activity of Antioxidant Enzymes

#### 2.2.1. Catalase (CAT) Activity

Table 4 compares the erythrocyte catalase (CAT) activities of HD patients from each group before and after HD in subsequent months of intake.

Mean CAT activities were statistically significantly different between almost all groups, except for Group 0 versus I and V, and Group II versus IV before and after HD regardless of the month of determination. Group III had the highest CAT activity, significantly higher than the other groups, in all months, before and after HD. Lower values were obtained in Groups IV and II, followed by I and 0, and the lowest was recorded in Group V in all months of the collection, before and after HD. In addition, the level of CAT activity before HD in Group I was significantly lower than in Group III, and similar differences were obtained between Groups II and III (II < III) and between III and IV (III > IV). There were significant differences in CAT activity after HD between Groups I and III (I < III), I and IV (I < IV), and II and III (II < III) in all months of intake. In addition, in Group 0, the enzyme activity was significantly lower than in Groups II, III, and IV both before and after HD, regardless of the month of the collection.

CAT activity increased in the following months of the study in almost all patient groups except for Groups 0 and V. Compared with baseline values, a statistically significant increase in CAT activity was observed in Groups I–IV. In month 12, activity was significantly higher than in month 3 and month 6 of intake in these groups. CAT activity in Group 0 was statistically significantly lower than in Groups II, III, and IV in subsequent examinations, while in Group V, it was not significantly different between blood sampling.

Hemodialysis did not cause statistically significant changes in CAT activity in the groups in all months of the assays although there was a slight increase in activity after HD in all cases.

One-year dietary therapy has been shown to affect catalase activity. The use of a standard diet enriched with natural antioxidants (Group II) raised CAT activity more effectively than the use of a standard diet (Group I). The best results were obtained when the diet was used in Groups III and IV.

#### 2.2.2. Superoxide Dismutase (SOD) Activity

A comparison of superoxide dismutase (SOD) activity in erythrocytes of HD-treated CRF patients from each group before and after HD in subsequent months of intake is shown in Table 5.

The mean SOD activities were statistically significantly different between almost all groups. There were no statistically significant differences between III vs. IV and V vs. I and II. The highest SOD activity was obtained in Groups III and IV in all months of determination, before and after HD. Lower values were found in Groups II, I, and V, while the lowest values were found in Group 0, regardless of the month of the collection, in blood before and after HD.

SOD activity increased in successive months in all patient groups except for Group 0. The mean SOD activities were statistically significantly different between the months of the collection; however, no statistically significant difference was found between the activity in months 6 and 12. The lowest SOD activity was obtained at the beginning of the study (month 0), higher in month 3, and the highest in months 6 and 12 in all groups except for Group 0. In this group, a decrease in enzyme activity was observed in the subsequent months of the study.

After hemodialysis, SOD activity statistically significantly increased in all groups in each month compared with values obtained before dialysis.

Patients in Group 0 showed a decrease in SOD activity during the 6 months of the study. Group II patients showed higher SOD activity compared with standard dieters (Group I) and those with hepatitis C (Group V). The addition of pharmacological supplements provided higher enzyme activity in HD patients.

#### 2.2.3. Glutathione Peroxidase (GSH-Px) Activity

Groups IV and III were characterized by high glutathione peroxidase activity in each month of determination, before and after HD. Lower values were obtained in Groups II, I, and V, with the lowest in Group 0 regardless of the month of the collection, before and after HD (Table 6).

Mean GSH-Px activity was statistically significantly different between almost all groups. There were no statistically significant differences between Groups I vs. II and V, and II vs. V.

GSH-Px activity increased in the following month in all patient groups except for Group 0. The mean GSH-Px activity was statistically significantly different in almost every month of intake in Groups I, II, III, and IV. No statistically significant difference was observed between the activity in months 3 and 6. The lowest values were obtained at the beginning of the study, higher values in months 3 and 6, and the highest values in month 12, except for Group 0. In contrast, Group 0 showed a systematic decrease in GSH-Px activity during the 6 months of the study.

Dialysis caused a slight, not statistically significant, increase in mean GSH-Px activity in all groups regardless of the month of intake.

The “standard” diet with pharmacological supplements (Group III) and the “modified” diet (Group II) were shown to have a beneficial effect on the GSH-Px activity of HD patients. An increase in enzyme activity was also provided by the “standard” diet used by HD patients in Groups I and V, although to a lesser extent. The use of the diet by patients in Group 0 decreased the enzyme activity.

### 2.3. Malondialdehyde (MDA) Concentration

Table 7 shows the comparison of plasma MDA concentrations of HD subjects between the groups before and after HD in the following months of intake.

The plasma MDA contents of HD patients were statistically significantly different between almost all groups. No significant differences were noted between Groups 0 vs. I and V, and between Groups I vs. V and II vs. III. The highest mean MDA concentration occurred in Group 0 and Groups I and V, lower MDA content was found in Groups II and III, and the lowest in Group IV. The plasma MDA content decreased during the study in Groups II, III, and IV for one year while it increased in Group I. In Group V, the concentration of MDA increased from baseline in the third month of the study and then decreased in the following months to values lower than the initial ones. In Group 0, on the other hand, the amount of MDA in patients increased in subsequent months of the study.

Mean MDA concentrations were statistically significantly different in almost all months of intake. Only between intakes in months 0 vs. 3 and between months 6 vs. 12, there were no statistically significant differences between groups. The lowest mean MDA concentrations were observed in months 12 and 6 of treatment and higher concentrations at 3 and 0 months in all groups. After 3 and 6 months of the study, lower MDA concentrations were observed in the other groups compared with Group 0. In the 12th month of the diet, the highest MDA content was observed in Group I.

Plasma MDA content before dialysis was statistically significantly higher than after dialysis in all groups, regardless of the month of determination.

It was shown that the diet in Group II was more effective in reducing lipid peroxidation than the diets in Group I, Group V, and Group 0. During the 12-month nutritional therapy, patients in Groups III and IV had the lowest MDA concentrations, which may indicate the beneficial effect of pharmacological supplements in reducing lipid oxidation.

### 2.4. Results of Determination of Cytokine Concentrations

#### 2.4.1. IL-1 Concentration

Table 8 shows the comparison of IL-1 cytokine [pg/mL] plasma levels of patients in different groups before and after HD before and after the study.

Before the start of the study, high IL-1 concentrations occurred in Groups III and I both before and after HD. Lower values were observed in Group IV and the lowest in Group II before and after HD. The use of a “standard” diet (Group I) and selected supplements (Groups III and IV) over 12 months resulted in a significant reduction in IL-1 cytokine in the plasma of dialysis patients.

A single hemodialysis treatment resulted in a statistically significant increase in the mean plasma IL-1 levels of patients in all groups regardless of the month of the study. Group III receiving supplements was the only group showing a decrease in the concentration of this cytokine when comparing the condition after and before HD.

The use of a standard diet with pharmacological supplements (Group III) was shown to most effectively reduce IL-1 release.

#### 2.4.2. IL-6 Concentration

A comparison of IL-6 [pg/mL] plasma levels in patients in different groups before and after one year of the study is shown in Table 9.

At the beginning of the study, the highest IL-6 values were obtained in Group I, with lower values in Groups IV, III, and II in samples before HD regardless of the month of the collection. However, after HD, the highest values were observed in Groups I, IV, II, and III. After one year of dietary intake, there was a significant decrease in IL-6 concentration in Group I before HD. The concentration of this cytokine also decreased in Groups III and IV. In contrast, an increase in IL-6 was observed in Group II compared with the initial values. These differences were not statistically significant.

There were significant differences in IL-6 concentrations between Groups I vs. II, III, and IV, and between III and IV, before and after HD at the start of the study.

The month of sampling significantly affected plasma IL-6 concentrations. Mean IL-6 contents were statistically significantly different between concentrations at 0 and 12 months in all groups before and after HD, with values higher in almost all groups at the beginning of the study. Only group II before HD showed an increase in IL-6 concentrations at 12 months.

The change in diet had a beneficial effect on interleukin-6 release after 156 dialysis treatments in all groups. Compared with Group I, reduced IL-6 release was demonstrated by the “modified” diet (Group II). The addition of pharmacological supplements in Groups III and IV was more effective in reducing IL-6 secretion than dietary supplementation with natural antioxidants. The best results were obtained in Group IV receiving selenium.

#### 2.4.3. IL-8 Concentration

Table 10 compares the plasma levels of cytokine IL-8 [pg/mL] of patients in different groups before and after HD at the beginning and end of the one-year study.

Before the start of the study, the lowest values of IL-8 concentrations occurred in Group II both before and after HD. Groups III and IV showed higher levels of interleukin, with the highest in Group I before and after HD. After 12 months of dialysis, the lowest IL-8 levels were observed in Group II, higher levels in Group III, and slightly higher in Group IV and Group I before and after HD.

Statistically significant differences with decreasing trends were found in almost all groups except for Group II where values did not change.

The mean IL-8 content before dialysis was statistically significantly lower than after dialysis in all groups regardless of the month of intake.

Both the standard diet in Group I and the standard diet with supplements in Group III or selenium in Group IV reduced IL-8 release over 12 months.

#### 2.4.4. TNF-α Concentration

Blood TNF-α content as a result of one year of dialysis therapy and appropriate diets are shown in Table 11.

Mean TNF-α concentrations were statistically significantly different between all groups, except for Groups II and III. Group I had the highest initial cytokine TNF-α levels both before and after HD. A lower value was obtained in Groups II and III, and the lowest in Group IV. During the 12 months of the study, TNF-α content before HD decreased statistically significantly in Group I, while it remained unchanged in Groups II and III and increased statistically significantly in Group IV.

The cytokine content was found to be significantly higher in samples after HD compared to before HD regardless of the month. However, it can be assumed that this is a temporary increase as a renewed decrease in serum TNF-α levels was observed after 12 months before HD, except for Group IV.

It was shown that the use of diet in Groups II and III did not change the TNF-α content. The use of Se supplementation in Group IV during the 12 months of the study stimulated TNF-α secretion.

#### 2.4.5. IL-10 Concentration

Table 12 shows a comparison of IL-10 cytokine levels in the plasma of patients from the selected groups, before and after HD, and before and after the diets.

Before starting the diets, the highest IL-10 values before HD were observed in Group II, lower values in Group I, and the lowest values in Groups III and IV. After 12 months on the diets, a decrease in IL-10 levels was observed in Groups I and II before HD. In contrast, there was an increase in Groups III and IV in blood samples taken before HD.

Using the ANOVA test for factorial systems, significant differences in IL-10 concentrations before and after HD before the start of the study (month 0) were found between Groups I vs. III and IV and II vs. III and IV. In contrast, there were no statistically significant differences between Groups I vs. II.

The month of sampling significantly affected plasma IL-10 concentrations. Mean plasma IL-10 concentrations were statistically significantly different before and after the study. In Groups, I and II, the plasma cytokine content in month 0 was higher than the concentrations obtained in month 12 in the pre- and post-HD samples. In Groups III and IV, interleukin 10 concentrations in samples taken at the start of the study were lower both before and after HD than those obtained in month 12.

The hemodialysis procedure caused an increase in IL-10 release in all groups, regardless of the month of intake. However, this increase appears to be temporary as the use of diet resulted in a decrease in IL-10 concentrations before HD compared with the results obtained at the beginning of the study after HD. No such relationship was observed in Group III.

Thus, an increased release of IL-10 was shown in patients who followed a standard diet with supplements (Groups III and IV) over 12 months both before and after HD. In contrast, the use of a standard diet by patients in Group I and a diet by patients in Group II resulted in a statistically insignificant decrease in plasma levels of this cytokine over 12 months, before and after HD.

## 3. Discussion

End-stage renal disease (ESRD) is a major public health problem. The kidneys are known to play a key role in maintaining nutritional homeostasis. Therefore, people with ESRD undergoing hemodialysis are advised to modify their diet [33,34,35,36]. In particular, it is recommended to limit the intake of phosphate, potassium, and sodium to avoid elevated serum electrolytes and associated cardiovascular complications [37,38,39,40]. At the same time, clinical guidelines include recommendations for energy and protein intake to avoid malnutrition [33,34,35,36]. Implementing these recommendations in practice is very difficult for most patients. The difficulty lies in the need to provide the right amounts of energy and protein while limiting phosphorus. And we know that products with a high biological value are also rich in phosphorus. The need to reduce the intake of vegetables and fruit is due to the restriction of potassium and sodium in the diet. It is suggested that meat and vegetables should be soaked and then cooked in plenty of water, preferably twice. Meat and vegetable stock should not be consumed. Canned fruit has a lower potassium content but should be drained of syrup before eating. Soups should be avoided due to their high water, sodium, and potassium content. All these cooking procedures reduce the content of antioxidant vitamins and bio-elements, which in the long term can lead to deficiencies and, thus, hinder the neutralization of free radicals. Aspects of the nutrition of these patients negatively affect their quality of life. Dietary education of patients (and their families) to promote a healthy diet and adherence to dietary recommendations appears to be crucial [27,28,29,30,41].

Malnutrition is as common a complication found in dialysis patients as chronic inflammation. Symptoms of malnutrition occur in 28–54% of patients undergoing renal replacement therapy [23]. Furthermore, it is recognized as one of the factors leading to impaired immune defense mechanisms and increased risk of infection in CRF. Malnutrition is known to affect many elements of the immune system—including complement function, secretory immunoglobulin secretion, cytokine production, and phagocytic cell function [42,43].

Based on the BMI results, it can be concluded that a well-balanced diet can reduce weight loss. During the one-year diet therapy, no weight loss was observed in the adherent groups. In Group 0, a systematic decrease in BMI was observed, which was associated with weight loss. However, the absolute magnitude of this index remained normal at each stage of the study in all groups.

Several studies show a paradoxically better prognosis for overweight or obese hemodialysis patients compared with normal-weight or underweight patients [44,45,46].

Reports in the literature on BMI in this group of patients are divergent. Ghorbani et al. [24] studied 239 HD patients and found a BMI ≤ 18.5 in 8.8%, between 18.5 and 24.9 in 49.8% and > 25 in 41.4%. A similar percentage of patients in each BMI range was observed by Saglimbene et al., studying a group of 6701 HD patients [34]. In a review of the literature, Aoyagi et al. cited further findings demonstrating a systematic decrease in body weight of chronic dialysis patients over 10–15 years [47]. Different results, confirming those obtained in our study, were obtained by Louden et al. They showed that an adequate energy and protein supply in a group of dialysis patients can limit weight loss and even contribute to an increase in BMI [48].

Considering albumin and total protein concentrations as indicators of nutritional status, the results of our study are promising. The mean concentrations of albumin and total protein in the study subjects were within the middle values of the norm. Only Group 0 showed a decrease in their concentration to the lower limit of the norm observed. Low albumin and total protein concentrations are commonly observed in dialysis patients and are confirmed by numerous studies [49,50,51]. In our study, we found lower (but within normal limits) albumin levels in dialysis patients with hepatitis C with normal total protein levels. Similar results were observed by other authors [52]. This may be because elevated levels of γ-lactoglobulin are observed in patients with hepatitis C, which may affect normal total protein but not albumin concentrations.

It has been shown that mortality among dialysis patients with serum albumin levels below 35 g/L is more than four times higher than in patients with albumin levels above this value. Serum albumin levels below 3.0 g/L are accepted as critical values indicative of significant malnutrition [53]. Albumin concentrations are influenced not only by the amount of dietary protein intake but also by the hydration of the patient. Low serum albumin concentrations may be due to the patient’s hypohydration, and the results of many studies show an association between albumin concentrations and inflammation [54].

Inflammation is associated with chronic renal failure both in the pre-dialysis period and after the start of renal replacement therapy. Elevated inflammatory markers are found in 40–55% of hemodialysis patients [55]. The elevated levels of pro-inflammatory cytokines found in patients with CRF are due not only to impaired excretion of these substances by the failing kidneys but also to their increased production. Stimulators of the inflammatory response in patients with chronic renal failure include increasing uremic toxemia, oxidative stress, latent foci of chronic infection, contact with biocompatible materials during the dialysis procedure, or contamination of the dialysis fluid [56].

The study undertaken was to determine the effect of dietary modification and the use of supplements on the levels of some pro-inflammatory cytokines, IL-1β, IL-6, IL-8, and TNF-α, in these patients.

In our study, the lowest initial concentrations of IL-1β, IL-6, IL-8, and relatively high TNF-α were observed in Group II compared with the other groups. The use of a diet enriched with natural antioxidants for 12 months was the most effective in reducing TNF-α concentrations. Supplement use in Groups III and IV was also shown to have a beneficial effect on the content of other cytokines. Group I had the highest initial concentrations of pro-inflammatory cytokines. The use of a ‘standard’ diet intended for dialysis patients did not significantly reduce serum levels of selected cytokines in hemodialysis patients.

Borazan et al. [57] also failed to achieve a reduction in IL-6 and TNF-α cytokine secretion after three months of dietary therapy in HD patients. The diet offered to dialysis patients met the guidelines intended for these patients. The effect of membrane type on cytokine release was excluded because, as in our study, polysulfone membranes were also used there.

The suggestion that excess IL-6 may contribute to impaired protein metabolism highlights the importance of regulating inflammation in the context of patient nutrition. In addition, the increased release of IL-6, which has been attributed to the bioincompatible membrane, points to the need for further research into the optimization of dialysis methods and their impact on nutritional status and inflammation [58].

A 6-month supplementation with nutritional supplements (Aminomel Nephro, Elolipid, l-carnitine, and vitamin E) contributed to an increase in BMI but did not significantly affect albumin levels. In addition, a decrease in TNF-α secretion and a slight increase in IL-6 were observed [59].

Other studies have shown a beneficial effect of an energy–protein balanced diet, supply of n-3 fatty acids, and vitamin E on pro-inflammatory cytokine levels. A reduced release of cytokines was observed in patients adhering to dietary recommendations [60]. Another study confirms the positive effect of n-3 fatty acids at a dose of 2.4 mg per day for 2 months on plasma IL-6 and TNF-α levels in HD patients. This suggests that supplementation with n-3 fatty acids may slow the development of atherosclerotic lesions [61]. Other researchers have also shown a beneficial effect of a proper diet enriched with n-3 fatty acids at a dose of 3 g/day for 2 months on the levels of pro-inflammatory cytokines [62]. Reiter et al. cite further findings confirming the beneficial effect of α-tocopherol on reduced secretion of cytokines, including IL-6 and TNF-α [63]. The results of our study are interesting because increased TNF-α release was observed in patients using a supplement of pharmacological preparations in addition to the ‘standard’ diet. Explanation of the results obtained requires further research as no studies explaining the relationship between selenium supplementation and increased TNF-α were found in the available literature.

Our study confirmed that the hemodialysis procedure causes a significant increase in pro-inflammatory cytokines in the plasma of patients from all groups. Other authors in studies have also shown an increase in cytokine levels after dialysis [64], or a slight increase in IL-1β, IL-6, and TNF-α was observed after hemodialysis and a significant decrease in IL-8 in the plasma of patients dialyzed with polysulfone membranes [65,66].

The human body defends itself against excessive production of pro-inflammatory cytokines through increased secretion of anti-inflammatory cytokines. T lymphocytes are responsible for the secretion of cytokines. They release a range of pro- and anti-inflammatory cytokines that interact with other immune response cells. Th (helper, auxiliary) lymphocytes comprise two subpopulations, Th1 and Th2, which differ in function. Th1 lymphocytes support the cellular response by secreting interleukin IL-2 and interferon γ, while Th2 lymphocytes support the humoral response. Th2 cells produce the interleukins IL-4, IL-5, IL-10, and IL-13. IL-10 inhibits the cellular-type immune response and the inflammatory response.

IL-10 may represent a physiological mechanism for inhibiting inflammatory responses caused by the accumulation of the inflammatory cytokines IL-1, IL-6, and TNF-α during the dialysis procedure. In addition, it inhibits hydrogen peroxide release and nitric oxide synthesis [67].

Our research showed that the most effective increase in IL-10 levels was achieved by annual supplementation with both vitamin E and C complex and Zn and Se in Group III and selenium in Group IV. An increase in IL-10 levels over 12 months was observed in both groups. The use of a standard diet by Group I patients and a diet enriched with natural antioxidants by Group II patients resulted in a decrease in plasma levels of this cytokine over 12 months, both before and after HD.

Other studies have shown a decrease in plasma IL-10 levels in HD patients during 12 months of follow-up. The authors studied the effect of nutritional status on the amount of serum concentrations of selected cytokines. They used plasma albumin concentration as an indicator of nutritional status. The authors assumed that patients followed a diet specifically designed for dialysis patients. They observed a systematic increase in the plasma albumin levels of HD patients to the middle values of normal, with a concomitant decrease in IL-10 levels [68]. Compared with healthy subjects, IL-10 levels were higher in dialysis patients, which is consistent with the results of other authors [57,69].

Studies on the effect of diet type were conducted by Borazan et al. [57]. They observed significantly higher IL-10 levels in HD patients compared with healthy subjects both at the beginning of the study and after 3 months on a diet designed for dialysis patients. The change in diet resulted in a slight increase in IL-10 compared with the values obtained at the beginning of the study [57].

Oxidative stress is a disruption of the body’s prooxidant–antioxidant balance and a shift toward oxidative reactions. It arises both endogenously and exogenously during the production of reactive oxygen species and when there are insufficient defenses against them [70].

Red blood cells are characterized by a highly effective defense system that includes antioxidant enzymes such as CAT, SOD, and GSH-Px.

One of the key antioxidant enzymes involved in the breakdown of hydrogen peroxide is catalase (CAT). In our study, we obtained significantly different CAT activity depending on the group.

A “standard” diet enriched with supplements (vit. C, vit. E, Zn, and Se) was shown to cause the highest increase in CAT activity. A diet enriched with natural antioxidants (II) and supplemental Se (IV) was found to be better than the “standard” diet (I). Failure to follow the recommendations for a proper diet (I) resulted in a decrease in enzyme activity. HD patients with hepatitis C (V) were initially characterized by lower CAT activity; however, after one year of treatment, an increase in its activity was observed.

Similar to the results of our study, results were obtained using vitamin E supplementation (600 mg/day) for 14 weeks. The group receiving the antioxidant had elevated CAT activity compared with healthy subjects and HD-treated patients not receiving the supplement [71]. A meta-analysis of 24 studies involving 512 hemodialysis patients showed that a 2–6-month oral supply of vitamin E increased the activity of the enzymes CAT, SOD, and GPX [72].

In our study, erythrocyte superoxide dismutase activity was the highest in Groups III and IV. Group II had significantly higher SOD activity than Groups 0, I, and V but lower than Groups III and IV. The mean SOD activity significantly increased from one study to the next in all groups except for Group 0.

It is known that the constituents of SOD are zinc and copper, and their presence ensures the proper functioning of the enzyme. The highest SOD activity in the group receiving zinc is not surprising. The results testify to the significant effect of supplements and a proper diet enriched with natural antioxidants on the activity of the enzyme.

Some researchers have observed higher SOD activity in HD patients, which may suggest the body’s response to oxidative stress associated with chronic kidney disease and the dialysis process [73,74]. In contrast, other studies indicate lower SOD activity in these patients, which may be due to the body’s reduced ability to produce antioxidant enzymes [75].

Hemodialysis-stimulated granulocytes produce superoxide anion radicals, which are inactivated by SOD. The lower SOD activity in HD subjects is also explained by the increased presence of hydrogen peroxide. Because of reduced GSH-Px activity in erythrocytes, accumulated hydrogen peroxide can inhibit SOD activity. High levels of hydrogen peroxide result from neutrophil activation during hemodialysis or overproduction by erythrocytes [76].

Dialysis caused a significant increase in SOD activity in all groups regardless of the month of intake. This may indicate that the body is preparing to defend itself against free radicals, which are formed as a result of oxidative stress caused by hemodialysis. The results are consistent with reports by other authors [77]. Zwolinska et al. observed a decrease in SOD activity in a group of hemodialyzed children [75].

No data were found in the available literature regarding studies on the effect of dietary modification on SOD activity.

In our study, higher SOD activity was determined in patients who received a complex of vitamin C, vitamin E, Zn, and Se than in the other groups. The beneficial effect of vitamin C supplementation on SOD activity was confirmed by other studies [78]. The results of the study of the effect of vitamin C supplementation on plasma SOD activity by Washio et al. were different. None of the doses of vitamin C used (200 mg, 400 mg, 1000 mg) caused a significant increase in the activity of the enzyme. The authors explain this by the competition between SOD and vitamin C in reactions to inactivate superoxide anion radicals. It is known that the neutralization of anion radical by vitamin C occurs more slowly, and even high plasma concentrations of vitamin C will not significantly affect the activity of the enzyme and its role in free radical scavenging [79].

The results of other authors’ studies on the effect of vitamin E on SOD activity are consistent with those obtained in the present study. An increase in the activity of the enzyme was shown in HD patients who were given vitamin E 600 mg/day for 14 weeks [71]. Similar results were obtained by other researchers, who confirmed an increase in SOD activity both in a group of HD patients who received oral vitamin E (400 mg/each HD for 3 weeks) and in a group of patients on dialysis with vitamin E-enriched membranes [80].

Similar results were obtained by Eiselt et al. [78]. The use of vitamin E-infused dialysis membranes with simultaneous vitamin C supplementation (500 mg/after each HD for 12 weeks) in the HD patient group resulted in the highest increase in SOD activity. Lower activity was determined in the hemodialysis group using cellulose membranes with additional vitamin C infusion and in the dialysis group using cellulose membranes without additional modifications, the lowest with vitamin E-modified membranes. In addition, the authors showed a slight reduction in SOD activity after HD treatment. Daily vitamin C intake in the study groups was estimated at 50 mg/day.

The high SOD activity in the group of selenium-treated patients obtained in our study is not confirmed by the study of Adamowicz et al. [81]. True, they showed higher SOD activity in the plasma of Se-supplemented HD patients compared with the control group (HD patients), but this was a statistically insignificant increase.

Glutathione peroxidase is important in catalyzing the reduction reaction of hydrogen peroxide by glutathione, which is converted to an oxidized form. It is a selenium-dependent enzyme, so, as expected in our study, the highest GSH-Px activity was obtained in the Se-supplemented patient group. The use of an antioxidant complex in Group III increased enzyme activity. A diet enriched with natural antioxidants contained in food had a favorable effect on GSH-Px activity in the blood of HD patients causing an increase. Group II achieved statistically significantly higher enzyme activity than Group 0, which did not follow dietary recommendations. It should be added, however, that no significant differences were noted between the group eating a standard diet for dialysis patients and a diet enriched with natural antioxidants.

GSH-Px activity increased with the next month of intake in all patient groups except for Group 0. The lowest values were obtained at the beginning of the study, higher values in months 3 and 6, and the highest in month 12. Thus, proper diet, antioxidants in food, and pharmacological preparations contributed to the increase in GSH-Px activity.

In the available literature, we did not find the results of studies by other authors evaluating the effect of a diet enriched with natural antioxidants on GSH-Px activity. Most of the papers deal with the comparison of the enzyme activity between HD patients and healthy subjects. Many authors have reported lower [82,83] or higher [75] GSH-Px activity in HD patients, while others have shown no differences in enzyme activity in dialysis patients and healthy subjects [73,84].

In our study, the dialysis process caused a slight, not statistically significant, increase in mean GSH-Px activity in all groups regardless of the month of intake. The higher GSH-Px activity is associated with mobilization of the enzyme to remove hydrogen peroxide and lipid radicals formed as a result of hemodialysis-induced oxidative stress. Other studies have observed a decrease in GSH-Px activity after dialysis in children, explaining this as the inhibition of GSH-Px activity by uremic toxins [75]. Also, a study by Yavuz et al. [85] showed lower GSH-Px activity in dialysis patients compared with healthy subjects. In addition, it has been noted that the activity of the enzyme is affected by the type of membrane used during dialysis. The use of a polysulfone membrane significantly decreased GSH-Px activity after dialysis, more than the use of a hemophane membrane. In contrast, no significant differences in enzyme activity were observed before dialysis. The authors concluded that polysulfone membranes induce greater oxidative stress. The results are in contrast to previously cited studies [85].

Our study showed that the Se-supplemented group had the highest GSH-Px activity.

The first to study the effects of selenium on its plasma levels and peroxidase activity in erythrocytes were researchers from the Saint-Georges team [86]. They administered selenium orally to a group of 20 HD patients at a dose of 500 µg/day for three months, then reduced the dose to 200 µg/day for another three months. After three weeks, they observed a significant increase in plasma selenium levels compared with the control group. They also showed an increase in GSH-Px activity in the blood, which reached levels as in the control group after 12 weeks of supplementation.

Studies by other authors confirm these reports. Administration of 300 µg/day of Se to HD patients for 3 months resulted in an increase in GSH-Px activity after the first month of supplementation. The maximum activity was reached in the second month of selenium administration, followed by a decrease in the third month, but the activity was still higher compared with the control group [81].

In our study, GSH-Px activity was high in Group III. The available literature indicates a beneficial effect of supplementation with vitamin E [71] and vitamin C [78] on the activity of the enzyme. Higher GSH-Px activity was shown in the group of patients who received vitamin E at a dose of 600 mg/day for 14 weeks, and it was comparable to healthy subjects [71]. Lower doses of vitamin E (400 mg/after HD treatment for 3 weeks) did not significantly affect GSH-Px activity [87].

Since dietary changes and supplements increase the activity of antioxidant enzymes, it was decided to see if they reduce oxidative stress damage. The most commonly used marker of lipid peroxidation for this purpose is the measurement of plasma MDA levels. Its content was determined in the serum of patients in all groups before and after HD.

Our study showed that in patients taking a standard diet enriched with natural antioxidants (Group II), MDA levels steadily decreased over the 12 months of therapy. The concentration of MDA in this group was lower than in Group III using supplements in the form of vit. C+ vit. E+ Zn complex in addition to the standard diet. These were statistically insignificant differences. Group I had significantly higher MDA values compared with Group II. However, the amount of MDA in plasma was most effectively reduced by the addition of selenium alone (Group IV). The results of other authors’ studies confirm that HD patients are characterized by elevated plasma MDA levels compared with healthy subjects [88,89].

Supplementation with antioxidant minerals and vitamins affected blood MDA levels in HD patients. Oral supplementation with vitamin E at a dose of 400 mg for 3 months after each dialysis treatment reduced MDA and anti-LDL oxidized antibody levels [90]. Using the same dose of vitamin E for 3 weeks, a similar effect was obtained [80]. Administering vitamin E at a dose of 600 mg/day for 14 weeks reduced blood MDA levels in HD patients [71].

In another study, HD patients were given vitamin C at doses of 120 mg/3 times a week for 3 weeks and then increased the dose to 500 mg for another 3 weeks. They noted that the doses of vitamin used caused an increase in MDA levels. Three months after the end of supplementation, they measured MDA and found that MDA levels dropped to values lower than the initial results. The authors suggest that high doses of vitamin C exacerbate lipid peroxidation and suggest administering the antioxidant only to patients with a proven deficiency [91].

A study by Chao et al. [92] confirmed the beneficial effects of the vitamin C and E complex supplementation and vitamins C and E alone on lowering MDA levels in HD patients. They obtained better results when vitamin E alone was used at a dose of 400 mg, 3 times a week for 6 weeks. The achievements of Antoniadi et al. contradict the above findings [73]. They showed that long-term administration of α-tocopherol (500 mg/day for 12 months) to HD patients increased blood MDA levels and decreased erythrocyte SOD activity. Paradoxically, the authors obtained a prooxidant effect of vitamin E.

The effect of three-month supplementation of vitamin C (250 mg/day) and Zn (20 mg/day) on lipid peroxidation in HD patients was studied. Significant reductions in plasma MDA levels in patients were demonstrated [93]. It was noted that plasma MDA levels before dialysis were significantly higher than after dialysis. This may indicate that the product of lipid peroxidation, a small-molecule compound, is removed during hemodialysis [66,94].

## 4. Materials and Methods

### 4.1. Characteristics of Investigated Groups

The study included 57 hemodialysis patients (19 women and 38 men) aged 61.7 ± 11.9 years (women 62.1 ± 11.1 and men 61.5 ± 12.4) at the Department of Nephrology, Transplantology and Internal Medicine, Pomeranian Medical University in Szczecin, Dialysis Centre Fresenius Nephrocare branch in Stargard, Dialysis Station No. 7, and branch in Drawsko Pomorskie, Dialysis Station No. 44. The cause of renal failure was glomerulonephritis in 19 cases, polycystic kidney disease in 9, hypertension in 3, pyelonephritis in 2, other causes in 9 (e.g., gout, IgA nephropathy), and the cause was unknown in 15.

Patients underwent hemodialysis regularly for at least 3 months (3–79 months). The mean duration of dialysis was 25 months. All patients were dialyzed for 3–5 h three times a week using Fresenius machines (Homburg, Germany) with polysulfone dialyzers. Conventional bicarbonate dialysis fluid was used in all patients. Blood flow through the dialyzer was constant at an average of 250 mL/min and dialysis fluid flow at 500 mL/min.

Patients with diabetes mellitus, liver failure, respiratory failure, chronic inflammatory conditions, or using hormone replacement therapy were not eligible for the study. In addition, patients using supplementation with vitamins and antioxidant minerals and patients with drug use, compulsive smoking, and alcohol consumption were also excluded. Patients who had had a blood transfusion were excluded from the study. Patients diagnosed with a malignant process and those who had experienced any acute inflammatory disease in the six months preceding the study were also excluded.

For fear of worsening their condition, the patients took the medicines they had previously taken regularly, e.g., calcium carbonate and B vitamin complex. After HD, all received intravenous recombinant human erythropoietin (rHuEPO). The average weekly dose was 4000 IU. In addition to CKF, 80% of patients had one or more concomitant diseases, including hypertension, ischemic heart disease, viral hepatitis, gout, and osteoporosis.

During the study, patients were allowed to drop out at any stage. Forty-one patients completed the program. Table 13 shows the size of the groups in successive months of the study and the reasons for not completing the study.

All qualified patients were presented with the project objectives and all gave informed consent to participate in the study. The project was approved by the Bioethics Committee at the Pomeranian Medical University in Szczecin (no. BN-001/118/06).

### 4.2. Study Design

The research had the following stages:(1)In all patients, the diet was initially assessed based on weekly menus recorded on dialysis and non-dialysis days. The correctness of the diet was assessed, taking into account the extent to which the patients’ energy needs were covered and the proportion of proteins, fats, carbohydrates, vitamins, and elements. The values obtained were compared with the current dietary standards [36].(2)In all patients, nutritional status was initially determined using body mass index (BMI). Dry body weight (after dialysis) was used to calculate it. A range of 18.5–24.9 was considered normal [95].(3)Following the dietary assessment, nutritional education was provided to patients. Patients were provided with specially prepared handbooks containing a list of products recommended/unadvisable/forbidden in dialysis diets and substitutes for protein-rich foods. To meet the objectives of the present study, some of the diets were modified, taking into account the information previously obtained about the products preferred or disliked by the patients. Individual weekly menus were provided to the patients.

The patients were divided into groups receiving, respectively:

**Group 0**—unchanged diet, considered the optimal home diet. People in this group considered their diet sufficient for their needs and did not need the care of a dietician. Only blood was taken from patients in this group for periodic examinations.

**Group I**—standard diet used in patients on hemodialysis. The following dietary assumptions were made: energy: 30–35 kcal/kg/day, protein 1.0–1.2 g/body weight/day, fluid intake 500–800 mL/day (including those contained in products) plus urine excretion, mineral supply: sodium 1800–2500 mg/day, potassium 2000–2500 mg/day, calcium up to 2000 mg/day including supplementation of calcium preparations, phosphorus 1000–1400 mg/day, and vitamin C 75–90 mg/day [36].

**Group II**—the same standard diet as the first group, but enriched with antioxidants naturally contained in food. The menus included additional, substitutable portions of fruit: half a grapefruit, an orange, a mandarin, a kiwi, and vegetables rich in antioxidants: broccoli, spinach, and beetroot. Compote was replaced with juices: grapefruit, blackcurrant, orange. The changes met daily potassium intake standards.

**Group III**—standard diet as the first group, but enriched with supplements: vitamin C—100 mg/day for 6 months (Kutnowskie Zakłady Farmaceutyczne POLFA S. A., Kutno, Poland), vitamin E—300 mg/day for 6 months (Medana Pharma Terpol, Sieradz, Poland), selenium—200 μg/day 3 times a week after each hemodialysis for 6 weeks (Walmark, Warsaw, Poland), zinc—5 mg/day for 6 months (Zincas, ZCF Farmapol, Poznan, Poland),

**Group IV**—standard diet enriched only with a 300 μg selenium preparation (Walmark, Watsaw, Poland),

**Group V**—(patients with hepatitis C)—standard diet used in patients treated with HD.

Groups I–V were receptive to nutritional education and agreed to work with a dietician. During dialysis with each patient, the diet was analyzed individually. During the 12 months of the study, individual weekly menus were prepared for patients in these groups.

During the 12-month follow-up, blood was periodically drawn from each patient for biochemical tests (in months 0, 3, 6, 12).

### 4.3. Serum Preparation

Venous blood was collected from an established arteriovenous fistula, using potassium sodium vermilate (EDTA) as an anticoagulant in months 0, 3, 6, and 12 from the start of the diet immediately before and after the dialysis procedure. Samples were centrifuged at 2000× *g* at 4 °C for 15 min to separate serum from blood cells. The serum was separated into sterile microtubes and preserved for determination of MDA, cytokines. Erythrocytes, after the ‘sheepskin’ of leukocytes was pulled off, were washed three times with 0.9% NaCl solution and centrifuged for 10 min at 2500 g after each wash. The cells were separated into sterile microtubes and preserved for determination of SOD, CAT, and GSH-Px enzyme activities.

Serum and erythrocytes were stored under freezing conditions (−20 °C) until assays.

### 4.4. Biochemical Analysis

#### 4.4.1. Determination of Total Protein and Albumin in the Blood of HD Patients

Total protein and albumin concentrations were determined before HD using an RX-Imola clinical chemistry analyzer (Randox, Crumlin, UK). Total protein concentrations of 62–84 g/L and albumin concentrations of 35–50 g/L were taken as reference values.

The results of total protein and albumin concentrations allowed the assessment of the nutritional status of the patients.

#### 4.4.2. Determination of Antioxidant Enzyme Activity (SOD, CAT, GSH-Px) and Malondialdehyde (MDA) Concentration

Enzyme activity and MDA concentration were determined spectrophotometrically. The measurement was performed on a UV/VIS Lambda 20 spectrophotometer from Perkin Elmer. Chemical reagents were purchased from Sigma Aldrich (Saint Louis, MO, USA).

##### Determination of Catalase (CAT)

Catalase activity was determined by spectrophotometric method (according to Aebi), using the enzyme’s ability to degrade peroxides [96].

Hemolysate with a concentration of 5 g/dL Hb was diluted with 50 mM phosphate buffer, pH 7.0, in a ratio of 1:500. To a sample containing 2 mL of diluted hemolysate was added 1 mL of 30 mM H_2_O_2_ solution. The decrease in the concentration of H_2_O_2_ degraded by catalase against a blank sample (2 mL of hemolysate and 1 mL of phosphate buffer) was measured. The change in extinction over 30 s at 240 nm at 20 °C was recorded.

##### Determination of Superoxide Dismutase (SOD)

Superoxide dismutase activity was determined by spectrophotometric method (according to Misra and Fridovich), using the oxidation capacity of epinephrine to adenochrome [97].

Hemolysate with a concentration of 5 g/dL Hb was inflicted with 0.4 mL of a mixture of chloroform and ethanol (3:5, *v*/*v*) and 0.6 mL of distilled water to extract SOD. The whole mixture was shaken for 1 min and centrifuged at 5000× *g* for 3 min at 4 °C. To the extract, 2.85 mL of carbonate buffer (pH 10.2) was added and incubated for 3 min. Before reading, 0.1 mL of epinephrine was added to the mixture. The change in the extinction of the reaction solution over 15 min was measured against a blank containing 2.95 mL of buffer and 0.05 mL of SOD extract at 320 nm at 30 °C. The change in extinction over time was also measured for the control sample against carbonate buffer as a blank. The control sample contained 2.9 mL of carbonate buffer and 0.1 mL of epinephrine.

##### Determination of Glutathione Peroxidase (GSH-Px)

Glutathione peroxidase activity was determined using the reduction reaction of hydrogen peroxide to water involving reduced glutathione with the formation of glutathione disulfide [98].

Hemolysate with a concentration of 3 mgHb/mL was incubated for 5 min with a transformation reagent (KCN and K_3_[Fe(CN)_6_]) at room temperature. A total of 0.5 mL of hemolysate was inflicted with a reaction mixture consisting of 1200 μL phosphate buffer pH 7.0, 3 μL glutathione reductase (GSSG-reductase), 100 μL glutathione (GSH), and 100 μL NADPH. This was then incubated for 10 min at 37 °C. The reaction was initiated by the addition of 100 μL of tert-butyl hydroperoxide and the decrease in NADPH concentration was measured. Changes in extinction were recorded over 5 min against a blank containing all components except for NADPH at 340 nm at 37 °C.

##### Determination of Malondialdehyde (MDA)

Plasma malondialdehyde concentration was determined using a spectrophotometric method according to Rice-Evans et al. without chromogen extraction [99].

In a centrifuge tube, 1 mL each of plasma, reagent A (15% (*m*/*v*) solution of trichloroacetic acid (TCA) in 0.25-mol HCl solution), and reagent B (0.37% (*m*/*v*) solution of thiobarbituric acid (TBA) in 0.25-mol HCl solution) were mixed. Two control samples were prepared in parallel: 1 mL of H_2_O was added to one instead of plasma, and 1 mL of H_2_O was added to the other instead of solution B. A total of 20 μL of GSH and 20 μL of EDTA were added to the tubes. The samples were then heated in a water bath at 100 °C for 10 min. After cooling, they were centrifuged at 3000× *g* for 10 min at 20 °C. The extinction of the filtrate was then measured at 535 nm against the first control sample. The absorbance of the second control sample was subtracted from the values obtained.

#### 4.4.3. Determination of Serum Levels of Pro- and Anti-Inflammatory Cytokines

The concentration of pro- and anti-inflammatory cytokines in patients’ serum was determined by immunofluorescence using a flow cytometer (BD FACSan, Franklin Lakes, NJ, USA), using CBA kits (no. 551811, BD Biosciences, Franklin Lakes, NJ, USA) according to the manufacturer’s instructions. The kits contained the necessary reagents for the determinations.

The following pro-inflammatory cytokines were determined: interleukin-8 (IL-8), interleukin-1β (IL-1β), interleukin-6 (IL-6), and tumor necrosis factor (TNF-α). In addition, the concentration of the anti-inflammatory cytokine, interleukin-10 (IL-10), was examined.

The method used microbeads coated with specific antibodies directed against individual cytokines in the serum. After incubation with the sera, the beads were coated with the cytokines. Specific antibodies labeled with PE dye were applied and the fluorescence intensity of the individual beads was recorded. It was proportional to the cytokine concentration and was compared with the standard curve.

The minimum detectable interleukin concentrations were 3.6 pg/mL for IL-8, 7.2 pg/mL for IL-1β, 2.5 pg/mL for IL-6, 3.3 pg/mL for IL-10, and 3.7 pg/mL for TNF-α.

Due to the high cost of the kits, cytokine concentrations were determined in sera before and after HD in months 0 and 12 (before the start and after the end of the study) in groups I, II, III, and IV.

### 4.5. Statistical Analysis

All results were presented as arithmetic mean ± standard deviation (SD). The results obtained were subjected to statistical analysis. The type of ANOVA analysis used showed that the distributions of the obtained results were not normal, so non-parametric tests were used for the analysis. The Wilcoxon paired t-test was used to assess the significance of the differences in parameters determined in each group before and after HD at different time intervals. It was assumed that the studied quantities differed significantly when the probability level was *p* ≤ 0.05. Statistical calculations were performed using Statistica 7.1 software from Statsoft (Krakow, Poland).

## 5. Conclusions

The study shows that a properly selected diet can slow the buildup of cachexia, reduce the release of pro-inflammatory cytokines, and ensure the normal activity of antioxidant enzymes in the blood of hemodialysis patients. Taking into account the beneficial effect of antioxidant vitamins and bio-elements on enzyme activity and the concentration of pro- and anti-inflammatory cytokines, the indications for their supplementation can be considered reasonable.

## Figures and Tables

**Table 1 ijms-25-11036-t001:** Body mass index (BMI) [kg/m^2^] in patients in each group before diet and after 3, 6, and 12 months of dietary recommendations (mean ± SD).

	Month	0	3	6	12
Groups					
**0**		23.3 ± 3.8	23.3 ± 3.9	22.8 ± 4.3	-
**I**		25.8 ± 6.3 ^1^	25.9 ± 6.3 ^2^	26.2 ± 6.2 ^3^	26.9 ± 6.1 ^4^
**II**		25.4 ± 3.8	24.8 ± 3.9	25.4 ± 3.7	25.5 ± 3.6
**III**		24.3 ± 5.3	24.2 ± 5.2	24.0 ± 5.3	24.5 ± 5.9
**IV**		23.8 ± 1.8	23.9 ± 1.7	23.9 ± 1.7	24.1 ± 1.6
**V**		21.6 ± 3.7 ^1^	21.9 ± 3.9 ^2^	22.1 ± 4.0 ^3^	22.5 ± 4.0 ^4^

^1^, ^2^, ^3^, …—groups marked with the same numbers are statistically different from one another at *p* < 0.05. **0**, **I**, **II**, **III**, **IV**, **V**—signify the group of patients.

**Table 2 ijms-25-11036-t002:** Comparison of albumin [g/L] in the plasma of hemodialysis patients (mean ± SD).

	Month	0	3	6	12
Groups					
**0**		40.2 ± 2.9 ^1,2,3^	38.5 ± 4.2 ^1,2,3^	38.3 ± 4.1 ^1,2,3^	-
**I**		42.1 ± 4.5 ^4,5,6^	39.5 ± 1.2 ^4,5,6^	40.2 ± 1.3 ^4,5,6^	41.7 ± 3.7 ^4,5,6^
**II**		43.6 ± 4.0 ^1,4,7^	43.8 ± 4.0 ^1,4,7^	44.9 ± 4.2 ^1,4,7^	46.1 ± 3.6 ^1,4,7^
**III**		44.7 ± 1.5 ^2,5^	44.9 ± 0.8 ^2,5^	45.6 ± 1.0 ^2,5^	47.2 ± 0.7 ^2,5^
**IV**		45.4 ± 0.9 ^3,6,7,8^	46.3 ± 0.9 ^3,6,7,8^	50.3 ± 9.7 ^3,6,7,8^	47.6 ± 0.5 ^3,6,7,8^
**V**		43.5 ± 2.0 ^8^	42.7 ± 3.3 ^8^	42.8 ± 2.8 ^8^	43.8 ± 2.2 ^8^

^1^, ^2^, ^3^, …—groups marked with the same numbers are statistically different from one another at *p* < 0.05. **0**, **I**, **II**, **III**, **IV**, **V**—signify the group of patients.

**Table 3 ijms-25-11036-t003:** Comparison of total protein [g/L] in the plasma of HD patients (mean ± SD).

	Month	0	6	12
Groups				
**0**		64.4 ± 7.5 ^1^	61.8 ± 3.1	-
**I**		62.7 ± 3.2 ^2^	69.5 ± 3.1	74.0 ± 4.1
**II**		66.1 ± 4.8 ^1,3^	71.7 ± 4.1	75.7 ± 3.7
**III**		73.1 ± 1.9 ^1,2,3^	76.0 ± 1.9	78.3 ± 1.7
**IV**		75.8 ± 0.5 ^1,2,3^	77.4 ± 1.0	77.8 ± 0.4
**V**		75.3 ± 3.4 ^1,2,3^	71.0 ± 2.4	76.4 ± 3.7

^1^, ^2^, ^3^, …—groups marked with the same numbers are statistically different from one another at *p* < 0.05. **0**, **I**, **II**, **III**, **IV**, **V**—signify the group of patients.

**Table 4 ijms-25-11036-t004:** Comparison of catalase activity [A/g Hb] in erythrocytes between groups before and after HD by month (mean ± SD).

	Month	0		3		6		12	
Groups		Before HD	After HD	Before HD	After HD	Before HD	After HD	Before HD	After HD
**0**		246.5 ± 25.4	258.8 ± 36.4	235.1 ± 22.7	241.2 ± 8.1	226.6 ± 14.5	228.0 ± 4.2	-	-
**I**				240.9 ± 30.2	246.6 ± 11.1	248.5 ± 11.1	251.1 ± 6.8	252.2 ± 7.3	256.9 ± 7.3
**II**				255.5 ± 14.3	258.0 ± 15.7	257.4 ± 10.0	259.3 ± 15.4	263.9 ± 16.3	264.9 ± 7.0
**III**				266.9 ± 29.4	276.2 ± 9.9	273.9 ± 8.9	278.7 ± 9.8	274.9 ± 5.3	280.2 ± 11.0
**IV**				257.1 ± 13.2	270.9 ± 11.3	260.0 ± 13.1	267.8 ± 7.4	261.1 ± 18.2	271.2 ± 8.4
**V**				220.3 ± 8.6	224.5 ± 8.7	220.7 ± 15.2	226.1 ± 5.4	231.8 ± 2.5	239.0 ± 6.7

**0**, **I**, **II**, **III**, **IV**, **V**—signify the group of patients.

**Table 5 ijms-25-11036-t005:** Comparison of superoxide dismutase activity [U/g Hb] in erythrocytes between groups before and after HD by month (mean ± SD).

	Month	0		3		6		12	
Groups		Before HD	After HD	Before HD	After HD	Before HD	After HD	Before HD	After HD
**0**		1783.1 ± 128.6	1805.4 ± 79.9	1749.3 ± 63.0	1789.7 ± 48.3	1622.5 ± 88.8	1664.5 ± 102.4	-	-
**I**				1836.6 ± 76.9	1843.6 ± 45.4	1906.4 ± 39.5	1914.9 ± 11.6	1967.9 ± 27.8	1970.7 ± 17.7
**II**				1886.6 ± 98.0	1922.7 ± 97.7	1963.0 ± 46.6	1968.3 ± 31.2	1977.7 ± 18.0	1983.1 ± 42.0
**III**				2013.7 ± 49.7	2027.7 ± 34.2	2039.2 ± 33.6	2042.1 ± 16.2	2045.1 ± 20.7	2050.5 ± 13.8
**IV**				1989.3 ± 60.2	1997.1 ± 33.3	2004.4 ± 65.2	2013.6 ± 38.8	2019.5 ± 31.8	2038.0 ± 40.1
**V**				1830.0 ± 57.0	1900.7 ± 53.0	1897.8 ± 30.3	1954.8 ± 44.8	1963.2 ± 36.7	1984.8 ± 46.7

**0**, **I**, **II**, **III**, **IV**, **V**—signify the group of patients.

**Table 6 ijms-25-11036-t006:** Comparison of glutathione peroxidase activity [U/gHb] in erythrocytes between groups before and after HD by month (mean ± SD).

	Month	0		3		6		12	
Groups		Before HD	After HD	Before HD	After HD	Before HD	After HD	Before HD	After HD
**0**		8.4 ± 1.2	8.7 ± 1.1	8.2 ± 1.0	8.7 ± 0.6	7.4 ± 0.5	8.0 ± 0.3	-	-
**I**				8.9 ± 0.7	9.1 ± 0.7	9.1 ± 0.2	9.2 ± 0.3	9.2 ± 0.1	9.5 ± 0.5
**II**				9.2 ± 1.1	9.5 ± 0.6	9.4 ± 0.6	9.5 ± 0.7	9.5 ± 0.6	9.6 ± 0.5
**III**				10.8 ± 0.7	11.5 ± 0.5	11.2 ± 0.6	11.3 ± 0.4	11.2 ± 0.6	11.3 ± 0.5
**IV**				11.2 ± 0.9	12.1 ± 0.8	11.8 ± 0.5	11.8 ± 0.4	11.9 ± 0.3	12.2 ± 0.2
**V**				8.7 ± 0.9	9.2 ± 0.4	9.2 ± 0.6	9.6 ± 0.7	9.4 ± 0.1	9.8 ± 0.8

**0**, **I**, **II**, **III**, **IV**, **V**—signify the group of patients.

**Table 7 ijms-25-11036-t007:** Comparison of serum MDA [μmol/L] concentrations between the groups before and after HD in each month (mean ± SD).

	Month	0		3		6		12	
Groups		Before HD	After HD	Before HD	After HD	Before HD	After HD	Before HD	After HD
**0**		1.406 ± 0.276	1.108 ± 0.069	1.460 ± 0.177	1.287 ± 0.12	1.516 ± 0.034	1.473 ± 0.03	-	-
**I**		1.350 ± 0.184	1.135 ± 0.096	1.334 ± 0.069	1.261 ± 0.042	1.318 ± 0.058	1.295 ± 0.054	1.446 ± 0.024	1.377 ± 0.043
**II**		1.413 ± 0.148	1.196 ± 0.075	1.295 ± 0.054	1.204 ± 0.033	1.225 ± 0.015	1.159 ± 0.036	1.196 ± 0.033	1.135 ± 0.035
**III**		1.302 ± 0.163	1.156 ± 0.072	1.288 ± 0.049	1.234 ± 0.044	1.256 ± 0.032	1.215 ± 0.039	1.212 ± 0.047	1.168 ± 0.025
**IV**		1.408 ± 0.137	1.158 ± 0.066	1.224 ± 0.039	1.165 ± 0.041	1.119 ± 0.039	1.180 ± 0.170	1.104 ± 0.035	1.063 ± 0.044
**V**		1.409 ± 0.143	1.166 ± 0.114	1.449 ± 0.103	1.325 ± 0.065	1.374 ± 0.041	1.351 ± 0.019	1.310 ± 0.012	1.279 ± 0.029

**0**, **I**, **II**, **III**, **IV**, **V**—signify the group of patients.

**Table 8 ijms-25-11036-t008:** Comparison of IL-1 cytokine levels [pg/mL] in the plasma of patients from different groups before and after HD in months 0 and 12 (mean ± SD).

	Month	0		12	
Groups		Before HD	After HD	Before HD	After HD
**I**		8.13 ± 0.06	8.90 ± 0.00	8.05 ± 0.21	8.52 ± 0.35
**II**		7.53 ± 0.23	8.05 ± 0.23	7.77 ± 0.26	8.24 ± 0.26
**III**		8.44 ± 0.42	8.94 ± 0.28	7.51 ± 0.23	7.88 ± 0.25
**IV**		7.90 ± 0.29	8.63 ± 0.38	7.72 ± 0.28	8.26 ± 0.35

**I**, **II**, **III**, **IV**—signify the group of patients.

**Table 9 ijms-25-11036-t009:** Comparison of IL-6 cytokine levels [pg/mL] in the plasma of patients from different groups before and after HD in months 0 and 12 (mean ± SD).

	Month	0		12	
Groups		Before HD	After HD	Before HD	After HD
**I**		15.90 ± 1.84	18.94 ± 0.65	6.13 ± 1.82	7.67 ± 2.32
**II**		4.20 ± 1.81	6.48 ± 2.71	5.05 ± 1.63	6.26 ± 2.00
**III**		4.23 ± 2.42	5.02 ± 2.45	3.79 ± 0.98	4.81 ± 1.62
**IV**		6.25 ± 3.92	8.09 ± 3.86	4.80 ± 0.96	6.50 ± 1.55

**I**, **II**, **III**, **IV**—signify the group of patients.

**Table 10 ijms-25-11036-t010:** Comparison of IL-8 cytokine levels [pg/mL] in the plasma of patients from different groups before and after HD in months 0 and 12 (mean ± SD).

	Month	0		12	
Groups		Before HD	After HD	Before HD	After HD
**I**		28.83 ± 7.22	35.00 ± 7.27	15.30 ± 3.38	19.97 ± 5.25
**II**		11.45 ± 3.92	17.93 ± 5.46	11.08 ± 4.33	18.05 ± 4.58
**III**		15.42 ± 10.23	20.31 ± 9.72	12.51 ± 3.32	16.17 ± 5.05
**IV**		16.88 ± 7.13	23.84 ± 11.15	14.90 ± 3.17	18.90 ± 3.72

**I**, **II**, **III**, **IV**—signify the group of patients.

**Table 11 ijms-25-11036-t011:** Comparison of serum TNF-α cytokine levels [pg/mL] in patients from different groups before and after HD in months 0 and 12 (mean ± SD).

	Month	0		12	
Groups		Before HD	After HD	Before HD	After HD
**I**		5.53 ± 0.38	5.87 ± 0.58	4.28 ± 0.13	4.88 ± 0.36
**II**		4.33 ± 0.24	4.67 ± 0.36	4.12 ± 0.30	4.74 ± 0.29
**III**		4.16 ± 0.19	4.44 ± 0.15	4.23 ± 0.27	4.87 ± 0.21
**IV**		3.78 ± 0.04	4.10 ± 0.22	4.26 ± 0.24	4.59 ± 0.17

**I**, **II**, **III**, **IV**—signify the group of patients.

**Table 12 ijms-25-11036-t012:** Comparison of IL-10 cytokine levels [pg/mL] in the serum of patients from the different groups before and after HD in months 0 and 12 (mean ± SD).

	Month	0		12	
Groups		Before HD	After HD	Before HD	After HD
**I**		4.12 ± 0.22	4.54 ± 0.25	3.70 ± 0.23	4.78 ± 0.52
**II**		4.20 ± 0.00	4.55 ± 0.06	3.69 ± 0.36	4.39 ± 0.65
**III**		3.39 ± 0.44	3.98 ± 0.45	4.30 ± 0.26	5.14 ± 0.55
**IV**		3.36 ± 0.09	3.90 ± 0.21	3.80 ± 0.33	4.70 ± 0.26

**I**, **II**, **III**, **IV**—signify the group of patients.

**Table 13 ijms-25-11036-t013:** Group size in consecutive months of the study and reasons for not completing the study.

	Month	0	3	6	12	Cause
Groups						
**0**		6	5	3	0	1-transplant 5-died
**I**		7	7	6	6	1-bad health
**II**		20	19	18	14	1-resignation 5-transplant
**III**		9	9	9	9	
**IV**		9	9	9	9	
**V**		6	6	5	3	2-resignation 1-died
Total		57	55	50	41	

## Data Availability

No new data were created or analyzed in this study. Data sharing is not applicable to this article.

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
