# Peer review of "Effects of Diet and Supplements on Parameters of Oxidative Stress, Inflammation, and Antioxidant Mechanisms in Patients with Chronic Renal Failure Undergoing Hemodialysis"

_ijms, 2024, doi:10.3390/ijms252011036_

Round 1
Reviewer 1 Report
Comments and Suggestions for Authors
The manuscript entitled Effects of Diet and Supplements on oxidative stress parameters and Antioxidant Mechanisms in Patients with chronic renal failure undergoing Hemodialysis in which the authors aimed to evaluate the effects of a balanced diet, on some parameters of oxidative stress, immune response, and nutritional status in hemodialysis (HD) patients.
The idea of the manuscript is good and can be accepted for publication after major revision.
The authors must revise the manuscript according to the following comments.
Abstract
The current abstract is not coherent. I would suggest re-writing the whole abstract concisely.
Introduction
The title indicates Diet but there is not a single line about diet in the whole introduction.
Moreover the authors mentioned well balanced diet in the abstract what does they mean from this?
I would recommend the authors to start the introduction from Diet, its types, various type of probiotics, metabolites present in the diet and its role in treating various diseases then they can move on further.
For instance the following two articles are of great help to improvise the introduction. If the authors wants to cite them they can cite them.
https://doi.org/10.1080/19490976.2023.2297864 .
https://doi.org/10.3390/nu15132956
https://doi.org/10.4067/S0717-97072022000105445 .
There are some grammatical mistakes which should be removed.
Results
There are too many tables I would suggest to at least make 3-4 figures from the tables.
Discussion
It is very lengthy it should be reduced and Latest literature should be added.
Methodology is well designed.
Conclusion should be added as a separate paragraph.
Comments on the Quality of English Language
The manuscript entitled Effects of Diet and Supplements on oxidative stress parameters and Antioxidant Mechanisms in Patients with chronic renal failure undergoing Hemodialysis in which the authors aimed to evaluate the effects of a balanced diet, on some parameters of oxidative stress, immune response, and nutritional status in hemodialysis (HD) patients.
The idea of the manuscript is good and can be accepted for publication after major revision.
The authors must revise the manuscript according to the following comments.
Abstract
The current abstract is not coherent. I would suggest re-writing the whole abstract concisely.
Introduction
The title indicates Diet but there is not a single line about diet in the whole introduction.
Moreover the authors mentioned well balanced diet in the abstract what does they mean from this?
I would recommend the authors to start the introduction from Diet, its types, various type of probiotics, metabolites present in the diet and its role in treating various diseases then they can move on further.
For instance the following two articles are of great help to improvise the introduction. If the authors wants to cite them they can cite them.
https://doi.org/10.1080/19490976.2023.2297864 .
https://doi.org/10.3390/nu15132956
https://doi.org/10.4067/S0717-97072022000105445 .
There are some grammatical mistakes which should be removed.
Results
There are too many tables I would suggest to at least make 3-4 figures from the tables.
Discussion
It is very lengthy it should be reduced and Latest literature should be added.
Methodology is well designed.
Conclusion should be added as a separate paragraph.
Author Response
The authors would like to thank the Reviewers very much for their time and preparation of the review. All comments are very valuable to us. Although not all comments have been taken into account we are open to discussion and further cooperation. We have included responses to the comments in the file.

Reviewer 2 Report
Comments and Suggestions for Authors
Title :
Mention inflammation in the title, as the paper revolves around oxidative stress and inflammation.
Abstract :
Line 16 : You use the abbreviation HD without clarifying its meaning.
Line 20 : You can use just HD instead of writing Hemoedialysis, since you will provide the full form in line 16.
Line 20: You have used 57 patients; please clarify their sex.
Introduction :
Line 45 : You use the abbreviation RFTs without clarifying its meaning.
Line 50: Change the sentence « opposing enzyme systems » by « Antioxidant enzyme system »
From Line 64 to 66 : Cite a reference (s) of this paragraph « An individually tailored diet can slow the development of kidney damage 64 and prevent metabolic disorders that occur in the course of chronic kidney disease (hy- 65 perkalemia, hyperphosphatemia, hypocalcemia) ».
From Line 73 to 77 : Cite a reference (s) of this paragraph The consequences of nutritional deficiencies in dialysis patients are numerous. Complications can be reduced by proper nutrition. The diet should contain an adequate amount of calories, moreover, it should be quantitatively and qualitatively balanced. Vitamins and bio-elements are particularly important components.
Line 74 : Cite some nutritional deficiencies in dialysis patients.
From Line 78 to 80 : Cite the exact parameters measured such as CAT ; SOD …………………
You are discussing nutrition. It would be preferable to add a paragraph with examples of nutrients.
In the manuscript, you measured oxidative stress markers and anti-inflammatory markers, but you did not dedicate sufficient space to addressing these parameters in the introduction, even though the paper revolves around them.
Results :
Tables : Draw the tables of the manuscript with the same design and according to the journal's guidelines. Additionally, divide the cells for the month group into two: one for the meanings of 0, I, II, III, and the other for the meanings of 0 and 12.
The titles of the results should be modified by removing the word 'Results'. The title should mention the sample used and the patients. Do the same for the other tables.
Remove line 83 and 84.
You used symbols or numbers to indicate significance between groups ; please choose either symbols or letters, not both.
Table 1 : The symbol * is used for comparison, but you did not specify between which groups it applies.
Table 1 : In the title, the word 'therapy' is used, but it is not a therapy; it is a diet. Please change it.
Table 1 : Provide the value for group 0 at the 12-month. Do the same for the other tables.
2.1. Results of Anthropometric and Biochemical Parameters’ is the overall title for Tables 1, 2, and 3. Please add subtitles (e.g., 2.1.1) for each table. Do the same for the other tables.
Table 2 had to be placed below the results description. Do the same for the other tables.
The legend for Table 2 is incomplete. You did not mention whether the results are presented as mean ± SD. Do the same for the other tables.
In the results description of all tables, please add values in the text to facilitate reading for the lecturer.
The results descriptions are not clear.
Discussion :
Line 344 and 345 : Please provide a clearer explanation.
From line 345 to 358 : You provided one reference. Please add supplementary references.
From line 366 to 381 : In the text, you only compare your results with others. Additionally, please provide an explanation of your findings.
Line 483 : Please try to explain the changes in anti-inflammatory markers by the diet provided to the patients.
Given the relationship between NRF2 signaling pathway and oxidative stress, you have to discuss your findings by this signaling pathway. In addition ; connect the changes of NRF2 and oxidative with diet.
Materials and Methods :
In the study, you used only 57 patients. Which statistical test did you use to confirm whether this sample size is representative? Additionally, the sample consists of 19 women and 38 men, and as you know, parameters can be influenced by sex. How could you explain this?"
It is preferable to reorganize the study design in the form of a table or figure to facilitate reading."
Determination of antioxidant enzyme activity (SOD, CAT, GSH-Px) and 755
malondialdehyde (MDA) concentration: You need to provide more details on the measurements and the reagents used for each parameter. Do the same for the pro- and anti-inflammatory cytokines
Author Response

(The authors gave the same response as above.)

Round 2
Reviewer 1 Report
Comments and Suggestions for Authors
The manuscript has been revised but still the English language of the manuscript is not satisfactory. It should be revised via a native speaker. Then the manuscript can be considered for publication now.
Comments on the Quality of English LanguageThe manuscript has been revised but still the English language of the manuscript is not satisfactory. It should be revised via a native speaker. Then the manuscript can be considered for publication now.
Reviewer 2 Report
Comments and Suggestions for Authors
In this version of the article, “Effects of Diet and Supplements on parameters of stress, inflammation, Oxidative and Antioxidant Mechanisms in patients with Chronic Renal Failure Undergoing Hemodialysis.” We can see an acceptable evolution compared to the first version because it has become more structured with more explanation.
the authors have taken the reviewer's remarks and suggestions into consideration, which has positively impacted the quality and consistency of the article.
with this version, the article shows an excellent scientific level and represents an added value in the interested research topics
the article is accepted for me with this version